# Determinants of COVID-19 Vaccine Acceptance and Uptake in a Transborder Population at the Mexico–Guatemala Border, September–November 2021

**DOI:** 10.3390/ijerph19116919

**Published:** 2022-06-06

**Authors:** Ietza Bojorquez, René Leyva-Flores, César Rodríguez-Chávez, Carlos Hernández-Campos, Marcel Arévalo, Ricardo Cortés-Alcalá, Georgina Rodríguez-Elizondo, Sandra Ward, Rebecca Merrill, Alfonso Rodriguez-Lainz, Dianne Escotto, Nirma Bustamante

**Affiliations:** 1El Colegio de la Frontera Norte, Km. 18.5 Carretera Escenica Tijuana-Ensenada, Tijuana 22506, Baja California, Mexico; ietzabch@colef.mx (I.B.); crodriguez82@gmail.com (C.R.-C.); chernandez@colef.mx (C.H.-C.); 2Instituto Nacional de Salud Pública, Av. Universidad 655, Cuernavaca 62100, Morelos, Mexico; r.natoley@gmail.com; 3Facultad Latinoamericana de Ciencias Sociales-Guatemala, 3A Calle 4-44 Zona 10, Guatemala City 01010, Guatemala; marevalo@flacso.edu.gt; 4Secretaría de Salud, Lieja No. 7, Ciudad de Mexico 06600, Mexico; ricardo.cortes@salud.gob.mx (R.C.-A.); georgina.rodrigueze@salud.gob.mx (G.R.-E.); 5Centers for Disease Control and Prevention, 1600 Clifton Rd., Atlanta, GA 30333, USA; vsp3@cdc.gov (S.W.); xdf6@cdc.gov (R.M.); jqi3@cdc.gov (A.R.-L.); 6Centers for Disease Control and Prevention, Sierra Nevada 115, Ciudad de Mexico 11000, Mexico; ivr8@cdc.gov

**Keywords:** COVID-19, vaccination, vaccine acceptance, social determinants, international borders, Mexico–Guatemala Border

## Abstract

Assessing COVID-19 vaccination uptake of transborder populations is critical for informing public health policies. We conducted a probability (time-venue) survey of adults crossing from Mexico into Guatemala from September to November 2021, with the objective of describing COVID-19 vaccination status, willingness to get vaccinated, and associated factors. The main outcomes were receipt of ≥1 dose of a COVID-19 vaccine, being fully vaccinated, and willingness to get vaccinated. We assessed the association of outcomes with sociodemographic characteristics using logistic regressions. Of 6518 participants, 50.6% (95%CI 48.3,53.0) were vaccinated (at least one dose); 23.3% (95%CI 21.4,25.2) were unvaccinated but willing to get vaccinated, and 26.1% (95%CI 24.1,28.3) were unvaccinated and unwilling to get vaccinated. Those living in Mexico, independent of country of birth, had the highest proportion vaccinated. The main reason for unwillingness was fear of side effects of COVID-19 vaccines (47.7%, 95%CI 43.6,51.9). Education level was positively associated with the odds of partial and full vaccination as well as willingness to get vaccinated. People identified as Catholic had higher odds of getting vaccinated and being fully vaccinated than members of other religious groups or the non-religious. Further studies should explore barriers to vaccination among those willing to get vaccinated and the motives of the unwilling.

## 1. Introduction

Vaccination is a major public health tool in responding to the COVID-19 pandemic, but multiple studies have shown suboptimal levels of vaccine acceptance in countries all over the world [1,2,3]. Studies conducted on COVID-19 vaccine acceptance varied by sex, age, ethnicity, education, and income [3,4,5,6] prior to the availability of the vaccine, pointing to the importance of social and cultural determinants in defining people’s attitudes towards vaccination. Additionally, declining childhood vaccination rates across all vaccines around the world [7] and the recognition by the World Health Organization (WHO) of vaccine hesitancy as one of the top ten threats to global health in 2019 [8], provide a glimpse into the possibility that the uptake of COVID-19 vaccines might be hampered by people’s willingness to receive them, even when available.

Vaccination is especially relevant in international land border areas, where transborder movement is frequent and the populations of neighboring countries interact on a daily basis [9]. In the COVID-19 pandemic, some countries have established land border closures for infection containment. However, complete border closures are difficult to implement and can have severe economic and social impacts [10]. In most parts of Latin America, where land borders are ubiquitously porous, populations are likely to keep moving across borders as their daily living requires, even with border closures officially in place [11,12]. Given this situation, the vaccination of transborder populations is a major aspect of the COVID-19 public health response.

The Mexico–Guatemala border is a place of intense population movement, with over 600,000 registered crossings in 2019 [13], to which an unknown number of irregular crossings should be added. Historically, it has been a region of constant economic, social, and cultural exchange, and many Mexican and Guatemalan nationals living in the area cross the border to work, shop, or visit family and friends. Additionally, hundreds of thousands of migrants (including asylum seekers) enter Mexico each year through this border, and many of them are turned back to Guatemala by migration authorities [14]. People living in the border area sometimes make transnational use of public and private health services [15], highlighting the strong linkage between the needs and resources of both countries.

Given the implications of these population dynamics for public health, a survey evaluating health practices of transborder crossers and themes around the COVID-19 pandemic in mobile populations at the Mexico–Guatemala border was implemented from September to November 2021 to inform public health policy. This article aims to describe the vaccination status and willingness to get vaccinated, as well as associated factors, in adult border crossers during the COVID-19 pandemic. During the time frame of survey implementation, COVID-19 vaccines in both countries included the Pfizer–BioNTech, Moderna, AstraZeneca, and Sputnik K vaccines. The CansinoBIO and Sinovac COVID-19 vaccines were also available in Mexico, and neither country had a vaccination requirement at land points of entry.

## 2. Methods

### 2.1. Design and Sample

We conducted a survey with a probability, time-venue sampling design, from September to November 2021, of transborder crossers entering three ports of entry from Mexico into Guatemala: La Mesilla, in the department (equivalent to state) of Huehuetenango, and El Carmen and Tecún Umán, in the department of San Marcos of Guatemala.

The survey design followed the methodology employed by Mexico’s Survey of Migration in the Southern Border (Emif-Sur), which focuses on events of border crossings in each time period. All estimators refer to events. However, the term “participants” is used to ease comprehension. Sampling includes two stages: selection of time-venue points and selection of individuals within the time-venue points. The sampling framework for the first stage was stratified by day of the week, consisting of combinations of 8 h periods within each day (time component) and cities and points of data collection at each city (venue component). The points of data collection are those with high volumes of border movement, which include the international borders, bus terminals, and areas where taxis and other public transport congregate. Time-venue primary sampling units were selected with probability proportional to the number of crossings. As the Emif-Sur’s sample design is intended to estimate the number of crossings during a given time period; there is no predefined sample size. Implementation of this sampling strategy since 2004 has shown that selecting 100 time-venue points in a three-month period renders a sample size sufficient to obtain estimators with adequate precision [16]. We selected a first-stage sample of 106 time-venue points, distributed, for operative reasons, across two months of data collection.

For the second stage of sample selection, at each time-venue selected point, interviewers with previous experience working in Emif-Sur and with additional training specific to this survey approached all consecutive persons crossing an imaginary line at the points of implementation. Two members of each team requested verbal consent from all eligible participants and administered questionnaires from consenting individuals using tables, while a third member of each team counted the number of persons passing through the same imaginary line. The proportion of eligible persons of the total of persons passing through was the basis for calculation of sampling weights. For further details on sample selection, please refer to the Emif-Norte y Sur reference included in this article [16].

### 2.2. Participants

Participants in this survey were persons crossing the border from Mexico into Guatemala at the selected points, regardless of nationality. The eligibility criteria were: (1) age 18 years or older and (2) having crossed into Guatemala in the past 12 h. The questionnaire was only available in Spanish, so those unable to respond to the questionnaire in this language were excluded from the survey.

### 2.3. Variables

The survey questionnaire included sections on nationality and place of living, sociodemographic variables, frequency and motives for border crossing, health status and use of health services, COVID-19 knowledge and perceptions, and COVID-19 vaccination practices (The survey questionnaire is available upon request). Given the rapid mobility of the population, the survey was limited to 10 min and included mostly multiple-choice questions and just a limited number of open-ended ones.

We evaluated three primary outcomes: vaccination status, completion of vaccination schedule, and willingness to get vaccinated. Vaccination status (yes/no response to the question “Have you been vaccinated against COVID-19?”) was asked of all participants through the survey period. From 15 October 2021 to 15 November 2021, we added questions regarding the type and number of vaccines received, so we were able to assess whether participants interviewed in that period were fully vaccinated. Fully vaccinated is defined as having received all recommended doses in a primary series of COVID-19 vaccine; two doses of those vaccines requiring a two -dose primary series or one dose for a single-dose primary series vaccine. Willingness to get vaccinated was evaluated by asking non-vaccinated participants if they would accept a free COVID-19 vaccine if they were offered one (yes/no answer). Additionally, a categorical vaccination status variable was constructed by dividing the sample in three groups: vaccinated (with at least one dose), unvaccinated but willing to get vaccinated, and unvaccinated and not willing to receive a vaccine.

As variables that could be associated with the outcomes, we considered country of birth and residence, sex, age, education level, ethnicity, religion, and most trusted source of COVID-19 information. We categorized country of birth and residence as born and living in Guatemala, born in Guatemala and living in Mexico, born and living in Mexico, and other. Sex was a dichotomous (male/female) variable registered as observed by the interviewer. Age in years was included as a count variable in models and categorized for descriptive purposes. Education level was categorized as: none, primary (1–6 years), lower secondary (7–9 years), upper secondary (10–12 years), and some college or more (>12 years). Ethnicity was evaluated with two questions, as is done in the Mexican census. The first question asks if the person self-ascribes as indigenous and the second if they self-ascribe as afrodescendant. We combined both questions into the following categories: indigenous, afrodescendant, both, or none of the above. We categorized religion as: none (without religious beliefs), spiritual without religious affiliation, Catholic, and other (a category including non-Catholic Christian denominations, Judaism, Buddhism, and Islam). Most trusted source of COVID-19 information was categorized as: Ministry of Health (in Guatemala or Mexico), news agencies, family or friends, and other. These variables are social and cultural determinants that have shown to influence COVID-19 vaccine acceptance, and they might also be associated with the opportunity to receive a vaccine (vaccine access). We hypothesized that a higher education level would have a positive association with uptake and acceptance. For all other variables, the results of previous studies have shown different directions for the association, so our hypotheses were bi-directional.

### 2.4. Analysis

After conducting exploratory and descriptive analysis for each variable, we explored the bivariate associations between the independent and dependent variables.

Next, we employed logistic regression models to evaluate the adjusted, multivariate association between the independent variables and the three primary outcomes. The first model, with vaccination status as a binary dependent variable, included all survey participants. The second modeled the association of independent variables with being fully vaccinated (versus having an incomplete vaccination schedule) and included only vaccinated participants for whom data on the type of vaccine and number of doses they had received were available. The third model had willingness to get vaccinated as the dependent variable and included unvaccinated participants. For each model, we followed a theory-driven approach, including all independent variables. We tested the models’ specification with the Archer–Lemeshow goodness-of-fit test and the link test. Dropping the variable for source of COVID-19 information significantly improved all three models’ fit, so that variable was not included in the final models.

All analyses considered the sampling design (weights, strata, and clusters) for the calculation of estimators and standard errors and were conducted using Stata (version 17, StataCorp, College Station, TX, USA, 2021).

### 2.5. IRB Approval

The protocol for the survey was reviewed and approved by the ethics committee of El Colegio de la Frontera Norte [079_230821]. This activity was deemed not to be research as defined in 45 CFR 46.102(l) and an IRB review was not required by the U.S. Centers for Disease Control and Prevention. Participants were informed of the aims and procedures and gave informed oral consent before responding to the questionnaire. No personal, identifiable data were collected from participants as part of this survey.

## 3. Results

During the two-month survey implementation period, we approached 23,710 persons, of whom 10,487 (44.2%) were ineligible (1018 because they had not crossed the border; 225 were ineligible because they were minors, and 9244 because they had not crossed the border in the past 24 h). A further 132 (0.6%) did not provide information to assess eligibility, and 5801 (24.5%) did not consent to participate. The remaining 7290 (30.7% of those approached) eligible and consenting participants, after applying the sample weights, represented an estimated total of 167,511 border crossings during the survey period. Of those eligible and consenting participants, 6518 (89.4%), provided data on vaccination status.

In Table 1, we describe the main sociodemographic and vaccination-related characteristics of participants. The mean age was 35.8 years, 33.6% were female, 38.9% indigenous, 0.7% afrodescendant, and 0.3% both indigenous and afrodescendant, and most had received education up to the primary level (57.3%). The majority (86.7%) were born and living in Guatemala; 6.2% were born and living in Mexico, and 3.2% were born in Guatemala and living in Mexico. Of the remaining 4.0% who were either from or living in other countries, most were Central Americans living in their country of birth or other neighboring countries. Most participants (56.0%) crossed the border at least once a month. Among the participants, 4.0% had been returned to Guatemala by the Mexican migration authorities (data not shown). As for religion, 11.8% of participants were non-religious; 6.3% were spiritual without a religious affiliation; 39.5% were Catholic, and 42.4% reported other religious denomination. This last category was composed mainly of non-Catholic Christian denominations (99.8%), with the remaining 0.2% being a combination of persons ascribing to Judaism, Buddhism, or Islam. The majority of participants stated that news agencies (74.6%) were their most trusted source of COVID-19 information, while only 20.2% reported the Ministry of Health (Guatemala or Mexico) as their primary choice.

Over half (50.6%) of participants included in the analysis sample reported having received at least one dose of a COVID-19 vaccine (Table 1). Of participants surveyed between 15 October 2021 and 15 November 2021, we estimated that 77.9% of those reporting having at least one dose of a COVID-19 vaccine were fully vaccinated. Of vaccinated participants living in Guatemala, 80.6% had received their vaccine in that country, and 97.1% of those living in Mexico had received their vaccine in Mexico. At the same time, 19.3% of those living in Guatemala had received their vaccine in Mexico, and 2.9% of those in Mexico had received their vaccine in Guatemala (not shown in Tables). The types of vaccine reported as received were Moderna (40.6%, CI 95% 35.2,46.1), AstraZeneca (19.8%, CI 95% 16.6,23.6), Pfizer (12.5, CI 95% 10.3,15.1), Sputnik (11.9%, CI 95% 8.8,15.9), CanSino (3.9%, CI 95% 2.7,5.6), and Sinovac (3.1%, CI 95% 2.3,4.1), while 8.2% of participants did not know which type of vaccine they had received (not shown in Tables).

Among the unvaccinated participants (*n* = 3162), 47.1% were willing to receive a vaccine (Table 2). Reported reasons for not being willing to receive a vaccine were fear of the side effects of either COVID-19 vaccines (47.7%) or vaccines in general (20.1%) and lack of trust in vaccines in general (16.8%). Religious beliefs and lack of trust in the government (as a general concept) were mentioned less frequently by participants as a reason for not being willing to receive a COVID-19 vaccine.

In Table 3, we show the distribution of categorical vaccination (vaccinated, unvaccinated and willing to receive a vaccine, or unvaccinated and not willing to receive a vaccine) by participants’ characteristics. The lowest proportion of unvaccinated willing to receive a vaccine was observed in persons ages ≥65 years. In addition, there was an education-related gradient in the prevalence of vaccination, even though only some differences were significant. As an example, 70.7% of participants with at least some college education had already been vaccinated, in comparison with only 43.1% of those with no formal education, and the latter had the highest proportion of participants not vaccinated and not willing to receive a vaccine.

Participants living in Mexico (including Guatemalans living in Mexico) had the highest proportions of vaccination, while Guatemalans living in Guatemala had the highest proportion of participants not vaccinated but willing to receive a vaccine. Daily border crossers had both the lowest proportion of vaccinated participants and the highest proportion of participants unvaccinated and unwilling to receive a vaccine in the future.

There were also differences by religious affiliation and the most trusted source of information. Participants who identified as Catholic had the highest proportion of vaccination. Those who reported no religious beliefs had the highest proportion of unvaccinated and unwilling to receive a vaccine. Participants who identified as non-Catholic Christians and members of other religious denominations had a lower proportion of vaccinated as compared to Catholic participants and a higher proportion of unvaccinated and not willing to get vaccinated in comparison with Catholic participants. Vaccination coverage was higher among those who chose the Ministry of Health as their most trusted source compared to media or family or friends, and the percentage of those not vaccinated and not willing to receive a vaccine was highest among the participants whose most trusted source was family or friends.

The first multivariate model (Table 4, Model 1) showed that after adjusting for other variables, people living in Mexico were more likely to be vaccinated, independent of their country of birth. The association of vaccination with education was upheld in the adjusted model, and those with 7–9 years of education and over were significantly more likely to be vaccinated than those with zero years of schooling. Ethnicity had no significant association with the likelihood of being vaccinated. Participants who identified as Catholic were the religious group with the highest proportion of vaccination.

Among the vaccinated (Table 4, Model 2), factors independently associated with being fully vaccinated were older age, living in Mexico, having a college-level education, and being indigenous. Members of non-Catholic Christian groups and other denominations were less likely than Catholics to be fully vaccinated. Finally, among the unvaccinated (Table 4, Model 3), those with some education were more likely to be willing to receive a vaccine than those who had not attended school. Participants who defined themselves as spiritual were more likely than Catholics to be willing to receive a vaccine.

## 4. Discussion

Using a representative sample of border crossings from three cities on the Mexico–Guatemala border, we estimated that as of October–November 2021, 50.6% of adult border crossers into Mexico had received at least one dose of a COVID-19 vaccine, 77.9% of whom reported being fully vaccinated. By comparison, on 4 October 2021, Mexico’s Ministry of Health reported 72% of the adult population in Mexico was vaccinated, but in Chiapas, the state bordering the sites where our survey was conducted, the vaccination coverage was only 48% in the adult population, supporting the results observed in our survey. At the national level, 70% of those vaccinated in Mexico were fully vaccinated, a percentage similar to the one we found in our sample [17]. In Guatemala, as of 15 December 2021, the national reported vaccination coverage was 24%, with slightly higher coverage in the departments where the survey was conducted: 34.6% in San Marcos and 38.0% in Huehuetenango [18,19].

The difference in vaccination coverage between countries may be an indicator of vaccine availability, which might explain why the vaccination coverage in this sample was higher among border crossers living in Mexico, and the fact that almost one fifth (19.3%) of border crossers living in Guatemala had received their vaccine in Mexico. Those living in Mexico (including Guatemalans living in Mexico) were more likely to be vaccinated, and unvaccinated Guatemalans living in Guatemala were most likely to be willing to receive a vaccine, suggesting the importance of the supply-side factors in vaccination uptake. These findings also indicate the relevance of transborder health service utilization for public health and the connected nature of the health systems of countries that share busy land borders with mobile transborder populations. Specific barriers to vaccination in those willing to receive a vaccine will be explored in future work. It is also interesting to note that a previous study of influenza vaccination at the Mexico–United States border showed, like ours, that frequent border crossers were less likely to be vaccinated [9]. This could be a result of the dynamics of daily life of this population or other factors, and it is a finding worth studying further.

In a summary of studies in low- and middle-income countries, Solis-Arce et. al. [3] found an average COVID-19 vaccine acceptance of 80.3% (CI 95% 74.9,85.6). Most of the studies those authors reviewed were conducted at a time when COVID-19 vaccines had not yet been authorized for emergency use, so they asked about hypothetical acceptance in the event of vaccine availability. The survey described here was conducted when vaccine campaigns in Mexico and Guatemala had already started, so the data are not strictly comparable. Yet, if the number of participants who were already vaccinated were added to those who were not vaccinated but willing to receive a vaccine, it would amount to 73.9%. This percentage is close to the one reported in that same Solis Arce et. al. study for Colombia (75%), so it is possible that our results reflect the particular social context of Latin America. Similar to the results described above, other studies have found that the main reason given for COVID-19 vaccine hesitancy is perceived side effects [3,20,21]. Other authors have mentioned the importance of mistrust in vaccine safety and conspiracy theories about COVID-19 vaccines [2], but these reasons were not apparent in our survey.

Slightly less than half of those not yet vaccinated in our survey (47.1%) were willing to do so, while the rest were unwilling to receive a vaccine. Thus, in the unvaccinated group, there seems to be nearly an even split among those who had not had the opportunity to receive a vaccine and those who had but were reluctant to do so. These two groups would require different approaches to improve their vaccination rates, and it is therefore important to continue exploring their characteristics and motives [22].

As for the social determinants associated with vaccine uptake and acceptance, we found education and religion to be consistently associated with these outcomes. In the Solis-Arce et. al. study [3], the association of education with vaccine acceptance varied by country, but in the only Latin American country included (Colombia), the association was positive, as was in our survey. Similarly, a review by Moola et al. [6] found that higher socioeconomic status was associated with vaccine acceptance in low- and middle-income countries, a factor that would allow for more access to education and maybe more access to vaccines too.

The differences in vaccination and willingness to get vaccinated by religious affiliation were interesting findings in our survey. Even after adjusting for other variables, people who identified as Catholic were more likely to have received a COVID-19 vaccine than other groups, which might point to the exclusion of minority religious groups from social benefits such as vaccination campaigns, possible selection bias specific to the region of implementation, or to different levels of trust or willingness to abide by governmental recommendations between majority and minority groups. In this regard, similar to our survey, a recent survey in Venezuela reported that non-Catholic university students were more vaccine-hesitant than Catholics, and related this to the sustaining of conspiracy theories [22]. However, while religious fundamentalism may be associated with conspiracy theories [23], religious affiliation may not, which was the focus of our questionnaire. As for those reporting no religious affiliation, at least one previous study found that non-affiliated spiritual participants were more likely to sustain conspiracy theories and refuse vaccines [24], while we found that unvaccinated members of this group to be more willing to get vaccinated. A more in-depth, contextualized study would be required to understand the COVID-19 related beliefs of diverse religious groups and of the non-religious, among this transborder population.

There are several limitations of our survey. It was a survey of events of crossing, where the same person could be included more than once, and it was offered only to Spanish speakers, potentially having a bias against indigenous persons. However, we believe it is unlikely that in the relatively short period of time of the survey (two months), a person would have been willing to respond to the survey twice. Other limitations include the possibility of over-reporting of vaccination because of social desirability bias and the possibility that those who weren’t included in the survey or analysis sample were different from the ones included in terms of migration status, language, or other variables associated with vaccination, considering ineligibility and refusal rates. As for its strengths, the probability sample design makes the results more generalizable than those of non-probability surveys, especially those conducted through the internet, which in Mexico, like most parts of the world, tend to render samples biased towards the higher socioeconomic groups [20]. Another advantage was the timing, which gave us opportunity to assess vaccine uptake and vaccine acceptance simultaneously.

Future work focused on transborder populations should inquire about specific barriers to vaccination secondary to system challenges and accessibility as well as those focused on acceptability. Qualitative studies would be useful to understand the association between religion and vaccine acceptance and uptake and to identify the best messages and ways to prioritize those not willing to receive a vaccine. The introduction of a new vaccine requires the confluence of factors such as political will and strong leadership, funding and infrastructure, partnership between different agencies, and adequate communication strategies [25]. As new vaccines are developed, these factors need to be considered, especially in low- and middle-income settings where vaccines are essential given the limitations for treatment that sometimes prevail [26].

## 5. Conclusions

Assessing COVID-19 vaccine uptake and acceptance of transborder populations in different parts of the world can inform public health policies focused on these areas, diminishing the need for strict and prolonged border closures while mitigating cross-border transmission. The finding of a 50.6% vaccination coverage in this population highlights the importance of increasing efforts in this regard.

Our results help inform efforts to increase vaccination coverage of transborder populations. For example, vaccination or information about where to receive a vaccine provided at the border could increase the likelihood of uptake. As differences by education level point to the social inequity in vaccination coverage, and the association with religion highlighted the importance of cultural aspects, it is important that campaigns consider these factors. Finally, the transborder use of health services to get vaccinated highlights the necessity of the health systems of neighboring countries to respond in a coordinated manner to public health emergencies such as the COVID-19 pandemic.

## Figures and Tables

**Table 1 ijerph-19-06919-t001:** Sociodemographic and vaccine-related characteristics of participants, survey of transborder populations at the Mexico–Guatemala border.

	TOTAL ^1^ (*n* = 6518)	
Variable	% or Mean	95% Confidence Interval
Age (years)	35.8	35.3, 36.3
Female	33.6	31.6, 35.8
Ethnicity		
Indigenous	38.9	36.2, 41.6
Afrodescendant	0.7	0.4, 1.3
Indigenous and afrodescendant	0.3	0.1, 0.7
Other	60.1	57.3, 62.9
Education level		
No schooling	12.8	11.1, 14.7
Primary (≤6 years)	57.3	54.9, 59.6
Lower secondary (7–9 years)	22.5	20.7, 24.4
Upper secondary (10–12 years)	6.3	5.6, 7.0
Some college or more (>12 years)	1.2	0.9, 1.5
Country of birth/residence		
Born and living in Guatemala	86.7	85.2, 88.1
Born in Guatemala, living in Mexico	3.2	2.6, 3.8
Born and living in Mexico	6.2	5.1, 7.4
Other	4.0	3.3, 4.7
Frequency of border crossing		
Daily	3.8	2.9, 5.0
Less than daily, at least 1/month	56.0	53.0, 58.9
Less than 1/month	40.3	37.6, 42.9
Religion		
None	11.8	10.0, 14.0
Spiritual, without religious affiliation	6.3	4.8, 8.3
Catholic	39.5	37.2, 41.8
Non-Catholic Christian or another religious affiliation ^2^	42.4	39.3, 45.5
Trusted source of COVID-19 information		
Ministry of Health ^3^	20.2	18.3, 22.3
News agencies	74.6	71.8, 77.2
Family or friends	4.1	3.1, 5.3
Other	1.2	0.9, 1.5
Vaccination status		
Vaccinated (at least one dose)	50.6	48.3, 53.0
Unvaccinated—willing to receive vaccine	23.3	21.4, 25.2
Unvaccinated—not willing to receive vaccine	26.1	24.1, 28.3
Fully vaccinated ^4^	77.9	75.2, 80.3

^1^ Analysis sample (those with data on vaccination status), unweighted *n*, all other data are weighted, confidence intervals consider sampling design. ^2^ See the methods section for a description of this category. ^3^ Ministry of Health of either Guatemala or Mexico. ^4^ Among the vaccinated, data available only for the second month of data collection. Defined as one or two doses, depending on vaccine type.

**Table 2 ijerph-19-06919-t002:** Willingness to receive vaccine and reasons for unwillingness, among the unvaccinated.

	TOTAL (*n* = 3162) ^1^	
Variable	%	95% Confidence Interval
Willing to receive vaccine	47.1	43.9, 50.3
Reasons for unwillingness ^2^		
Doesn’t trust the COVID-19 vaccine	8.9	6.7, 11.7
Doesn’t trust vaccines	16.8	13.7, 20.4
Fear of side effects of the COVID-19 vaccine	47.7	43.6, 51.9
Fear of side effects of vaccines	20.1	16.4, 24.4
Religious beliefs	3.5	2.3, 5.2
Lack of trust in government	2.9	1.8, 4.5
Other	0.1	0.0, 0.3

^1^ Unvaccinated, unweighted *n*, all other data are weighted, confidence intervals consider sampling design. ^2^ Among participants not willing to receive a vaccine. Mutually exclusive answers.

**Table 3 ijerph-19-06919-t003:** Sociodemographic characteristics of participants, by vaccination status and willingness to be vaccinated.

	Vaccinated		Not Vaccinated and Willing to Vaccinate		Not Vaccinated and Not Willing to Receive Vaccine	
Variable	% ^1^	95% Confidence Interval	% ^1^	95% Confidence Interval	% ^1^	95% Confidence Interval
Age (years)						
18–29	48.9	45.6, 52.3	27.9	25.1, 31.0	23.1	20.1, 26.5
30–64	51.5	48.8, 54.1	21.1	19.2, 23.4	27.4	25.3, 29.5
65+	50.7	37.9, 63.4	12.2	5.5, 25.2	37.0	25.8, 49.9
Sex						
Male	49.9	47.0, 52.9	24.5	22.2, 26.9	25.6	23.4, 27.9
Female	52.0	49.2, 54.7	20.9	18.2, 23.8	27.2	24.6, 30.0
Ethnicity						
Indigenous	47.2	42.7, 51.8	24.2	21.6, 27.0	28.6	24.5, 33.2
Afrodescendant	63.6	41.4, 81.2	13.3	4.7, 32.0	23.2	12.9, 38.0
Indigenous and afrodescendant	32.9	14.3, 58.9	31.6	11.5, 62.1	35.5	21.0, 53.3
None of the above	52.8	50.4, 55.1	22.7	20.2, 25.5	24.5	22.6, 26.4
Education level						
No schooling	43.1	36.7, 49.8	16.4	12.5, 21.1	40.5	35.1, 46.2
Primary (≤6 years)	46.2	43.6, 48.8	26.8	24.4, 29.4	27.0	24.8, 29.4
Lower secondary (7–9 years)	60.4	56.6, 64.1	21.1	18.2, 24.4	18.5	16.1, 21.1
Upper secondary (10–12 years)	67.8	62.6, 72.6	14.1	10.7, 18.4	18.1	13.8, 23.3
Some college or more (>12 years)	70.7	54.5, 84.0	14.1	6.9, 26.5	15.3	7.3, 29.4
Country of birth/residence						
Born and living in Guatemala	48.6	46.0, 51.2	24.8	22.8, 27.0	26.6	24.3, 28.9
Born in Guatemala, living in Mexico	62.6	53.2, 71.1	13.1	8.7, 19.3	24.4	16.8, 34.0
Born and living in Mexico	68.1	62.0, 73.6	10.5	7.1, 15.3	21.4	17.1, 26.5
Other	58.6	49.5, 67.2	16.4	12.1, 21.8	25.0	17.1, 35.0
Frequency of border crossing						
Daily	35.6	28.3, 43.7	22.7	14.5, 33.6	41.7	31.8, 52.4
Less than daily, at least 1/month	52.9	49.9, 56.0	25.3	22.6, 28.3	21.7	19.8, 23.8
Less than 1/month	46.5	42.2, 50.8	22.6	19.5, 26.0	30.9	26.4, 36.0
Religion						
None	44.3	39.6, 49.1	16.7	13.7, 20.3	39.0	34.6, 43.6
Spiritual, no religious affiliation	40.9	34.0, 48.3	46.0	39.0, 53.2	13.0	10.0, 16.8
Catholic	56.5	53.1, 59.8	22.2	19.4, 25.2	21.4	18.8, 24.2
Non-Catholic Christian or another	48.4	44.9, 52.0	22.7	20.2, 25.4	28.9	26.1, 31.9
Trusted source of COVID-19 information						
Ministry of Health ^2^	59.0	55.3, 62.7	15.3	12.6, 18.6	25.6	22.6, 29.0
News agencies	48.5	45.7, 51.4	25.8	23.6, 28.1	25.7	23.2, 28.3
Family or friends	44.5	37.9, 51.3	21.9	16.6, 28.3	33.6	27.5, 40.2
Other	66.0	51.9, 77.7	15.4	9.1, 24.8	18.7	10.8, 30.4

^1^ Weighted percentage, confidence intervals consider sampling design. ^2^ Ministry of Health of either Guatemala or Mexico.

**Table 4 ijerph-19-06919-t004:** Factors associated with vaccination and willingness to be vaccinated.

	Model 1: Odds of Vaccination ^1^			Model 2: Odds of Full Vaccination ^2^			Model 3: Odds of Willingness ^3^		
	Odds Ratio	95% Confidence Interval		Odds Ratio	95% Confidence Interval		Odds Ratio	95% Confidence Interval	
Sex (ref. male)	1.01	0.88	1.15	1.23	0.85	1.77	0.80	0.62	1.03
Age (yrs.)	1.00	0.99	1.01	1.03	1.02	1.05	0.99	0.98	1.00
Country of birth/residence									
Born and living in Guatemala	Ref.								
Born in Guatemala, living in Mexico	1.57	1.04	2.36	2.40	1.15	5.00	0.62	0.34	1.13
Born and living in Mexico	2.06	1.50	2.83	1.72	1.03	2.85	0.70	0.42	1.16
Other	1.33	0.85	2.06	1.11	0.57	2.15	0.62	0.35	1.08
Education level									
No schooling	Ref.								
Primary (≤6 years)	1.13	0.84	1.52	1.13	0.64	1.98	1.97	1.40	2.76
Lower secondary (7–9 years)	1.84	1.33	2.53	1.04	0.58	1.89	2.39	1.65	3.48
Upper secondary (10–12 years)	2.55	1.75	3.72	1.26	0.62	2.55	1.88	1.07	3.31
Some college or more (>12 years)	2.45	1.08	5.56	6.60	1.43	30.35	2.39	0.98	5.80
Ethnicity									
Not indigenous, nor afrodescendant	Ref.								
Indigenous	0.97	0.79	1.19	3.35	2.09	5.38	0.88	0.64	1.22
Afrodescendant	1.72	0.69	4.30	n.c. ^4^			0.56	0.21	1.52
Indigenous and afrodescendant	0.50	0.18	1.35	n.c.			0.74	0.23	2.42
Religion									
Catholic	Ref.								
None	0.58	0.45	0.74	0.77	0.48	1.23	0.41	0.31	0.55
Spiritual, without religious affiliation	0.59	0.44	0.80	1.50	0.98	2.31	3.07	2.10	4.50
Non-Catholic Christian or another	0.77	0.64	0.93	0.65	0.47	0.90	0.80	0.63	1.01

^1^ At least one dose of a COVID-19 vaccine, unweighted *n* = 6517, *p* < 0001. ^2^ Among the vaccinated, data available only for the second month of data collection, defined as one or two doses, depending of vaccine type, unweighted *n* = 1718, *p* < 0001. ^3^ Among the unvaccinated, unweighted *n* = 3162, *p* < 0001.^4^ n.c. = No cases.

## Data Availability

The data presented in this survey are available on request from the corresponding author. The data will be made publicly available once approval by the Mexican Ministry of Health is obtained.

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
