# Peer review of "Determinants of COVID-19 Vaccine Acceptance and Uptake in a Transborder Population at the Mexico–Guatemala Border, September–November 2021"

_ijerph, 2022, doi:10.3390/ijerph19116919_

Round 1
Reviewer 1 Report
Dear authors, the paper describer factors associated with vaccine acceptance and uptake in a transborder population at the Mexico-Guatemala border, mostly born and living in Guatemala.
The paper shows that a quite big proportion of the sample was unvaccinated and unwilling to get vaccinated. Factors associated with this choice were age, female sex, ethnicity, education and faith.
The main limitation regards the possible selection BIAS (only spanish -speaking individuals, persons could have been included more than once)
I think that the overall design, statistical analysis and discussion are well described, correctly pointing out the limitations of the study; based on the available data, the results presentation is informative.
I do not have suggestions to improve the quality of the study, because there is not the possibile to reduce the risk of Bias.
I could suggest to briefly discuss that the recent advances achieved in vaccine technology, that on one hand are a big success for the modern medicine, would be very important especially in the developing countries, but could be scary for the population (https://pubmed.ncbi.nlm.nih.gov/33374343/)
Author Response
We thank the Reviewer for his/her suggestion of reading DOI: 10.3390/vaccines9010009, and of discussing the importance of new vaccines for low- and middle-income countries. We added a reflection on this (lines 377-382).
Reviewer 2 Report
Thank you for the opportunity of reviewing this manuscript. The abstract states purpose, structure, and points made about the study.
There were some citations within five years, but the larger majority of references are older than five years
It also needs a statement provided by the statistician that the analysis is accurate based on the variables selected.
Author Response
The Reviewer observes that “...the larger majority of references are older than five years”. We respectfully disagree, since 23/25 references in the original manuscript are from 2018 or later (most of them from 2020 or later).The
Reviewer also asks us to add and “statement by the statistician that the analysis is accurate based on the variables selected”. We’re not sure where such a statement should be placed, since as far as we understand it, by submitting the article we are implying that the analysis is accurate. Nevertheless, we’re prepared to add such a statement if the Editors consider it is needed.
Reviewer 3 Report
Overall, I think this is an interesting, well-designed and executed research.
Below, I include my comments to each section of the article, and I suggest aspects of improvement in some of them. The comments to the authors are written in Spanish (it is the same information).
Authors' affiliation: Cuidad de Mexico (Ciudad)
Abstract: Describes the main sections of the research (introduction, objectives, methodology, results and conclusions). Since it is an unstructured summary, in my opinion it is not necessary to indicate RESULTS in capital letters, since the name of the rest of the sections does not appear.
Introduction: In my opinion, the section is complete and correct, including all the information necessary to understand the situation and the justification of the research. The objective is clearly described. I thought it would be useful to know the vaccination situation at the time the study was carried out in each country, but I have seen that it is included in the discussion section.
Methodology: The sampling strategy is described, replicating a proven methodology (Migration survey in the southern border of Mexico). Eligibility and exclusion criteria are described. Variables are described. The statistical analysis is described. Everything is correct in my opinion. The ethical aspects are described.
Results: The people who agreed to participate in the study and those who did not are described. All the necessary results are described based on the variables, the research objectives and the statistical analysis described in the methodology. It is correct.
Aspects for improvement: I think it would be appropriate to describe the 44.2% of ineligibles according to the specific reason (as indicated in the eligibility or exclusion criteria).
The tables cut paragraphs of text making reading difficult. It would be convenient to edit the presentation in the journal so that they appear at the end of the corresponding paragraph.
The column of variables in Tables 1, 2, 3 and 4 is center justified and without highlighting the names of categories with subcategories, making reading and interpretation difficult (e.g., ethnicity, educational level, etc.) in Table 1. I suggest left-justifying, bolding the categories, and indenting the subcategories so that they are better visualized and distinguished from the categories.
In Table 2, in addition, it would be convenient to have the entire variable name in one row and to align well with the percentages, as well as to highlight the two main categories and indent the subcategories of the second one.
Discussion: The results are discussed in terms of the research objectives and the scientific literature. The limitations of the research and proposed future directions are described. In my opinion it is complete and correct.
Conclusions: Part of the results are reflected.
Aspects for improvement: I think that some more paragraphs should be included in relation to the main objective of the research (% of vaccination). I think that the paragraph about the cross-border use of health services for vaccination conveys the opposite sense to the desired one (not that it undercores but that it highlights the need to respond in a coordinated manner).
Author Response
We have followed the Reviewer’s suggestions, as follows:
- We removed the word “RESULTS” from the abstract
- We described the reasons of ineligibility of the 44.2% ineligibles (lines 183-185)
- We moved the tables, so they don’t cut paragraphs of text
- We left-justified the column of variables in Tables 1,2,3 &4, and indented the sub-categories of variables
- In Table 2, we aligned the whole variable name in one row in line with the %, and indented the subcategories of the second variable
- We added a conclusion about the % of vaccination (lines 387, 388)
- We changed the word “underscore” to “highlight” to clarify the idea that cross border use of services makes it necessary to respond in a coordinated manner (line 394)